# Somali Information Retrieval Corpus: Bridging the Gap between Query Translation and Dedicated Language Resources

**Abdisalam Mahamed Badel[1], Ting Zhong[1], Wenxin Tai[1,2*], Fan Zhou[1,2]**

[1]University of Electronic Science and Technology of China, Chengdu, Sichuan, China
[2]Kashi Institute of Electronics and Information Industry, Kashi, Xinjiang, China
fiicane121@gmail.com, zhongting@uestc.edu.cn, wxtai@outlook.com,
fan.zhou@uestc.edu.cn

## Abstract

Despite the growing use of the Somali language in various online domains, research on Somali language information retrieval remains limited and primarily relies on query translation due to the lack of a dedicated corpus. To address this problem, we collaborated with language experts and natural language processing (NLP) researchers to create an annotated corpus for Somali information retrieval. This corpus comprises 2335 documents collected from various well-known online sites, such as hiiraan online, dhacdo net, and Somali poetry books. We explain how the corpus was constructed, and develop a Somali language information retrieval system using a pseudo-relevance feedback (PRF) query expansion technique on the corpus. Note that collecting such a data set for the low-resourced Somali language can help overcome NLP barriers, such as the lack of electronically available data sets. Which, if available, can enable the development of various NLP tools and applications such as question-answering and text classification. It also provides researchers with a valuable resource for investigating and developing new techniques and approaches for Somali.

## 1 INTRODUCTION

Information retrieval (IR) studies help design important techniques for extracting information from a large collection of data (Bhavadharani et al., 2019). There are several applications of IR in which search engine is the most known application. However, attempts made to develop such information retrieval applications for the Somali language are limited and primarily rely on query translation (Karakos et al., 2020; Boschee et al., 2019; Ogundepo et al., 2022).

Cross-language information retrieval has certain limitations, such as finding the appropriate translation of the source language query (Bhattacharya et al., 2016). Therefore, building the Somali information retrieval corpus will significantly contribute to the existing literature. This corpus will assist researchers in conducting further research on it or using it in other areas of NLP. Mainly, it can be used for text classification-related tasks and question-answering research purposes. So far, existing research on the Somali language has not provided a monolingual corpus for this language. The absence of essential digital resources is the key obstacle to advancing language technologies in various low-resourced languages (Babirye et al., 2022). Thus, building electronic data sets such as IR system evaluation corpus provides valuable support to the scientific community. Corpus as an electronic data set contributes extensively to the study of the language in many ways, such as syntax, semantics, and discourse (Rayson et al., 2006). An annotated corpus is a powerful tool for the evaluation and optimization of IR systems (Kazai et al., 2012).

In this paper, we created an IR corpus for the Somali language. The corpus is annotated manually by preparing a group of queries and relevant judgments used to evaluate the corpus based on the requirements in the Text Retrieval Conference (TREC) [1]. After the corpus was built, we evaluated it using the pseudo-relevance feedback (PRF) query expansion technique. This type of query expansion technique operates on the assumption that it retrieves first an initial set of top-ranked documents. Then, the most frequent words from the top-ranked documents are added to the previous query terms to modify the query, in our case, the top 5 terms. The newly formulated query terms are then submitted to the system to retrieve the relevant documents. The proposed approach for Somali information retrieval has shown a good performance.

**Contributions:** The main contributions of this paper are three-fold:

---

*Corresponding author: wxtai@outlook.com

[1]https://trec.nist.gov/data/test_coll.html

- We have created a corpus for Somali IR system evaluation.

- We proposed a PRF-based technique to evaluate the corpus.

- We identified a list of Somali stop words and used them as part of the research.

## 2 RELATED WORK

An essential tool for IR is the corpus or test collection, as introduced in (Scholer et al., 2016). By constructing a corpus, various algorithms can measure the system's effectiveness. An idea offered by (Kato et al., 2021), also strongly reveals the trend toward open data for scientific openness, (Kato et al., 2021) proposed a new test collection for ad-hoc retrieval, addressing issues such as techniques used and challenging topics. Furthermore, to create a corpus that accurately reflects real user behavior, (Kazai et al., 2012) found systematic judgment errors by comparing labels assigned by different individuals. (Kazai et al., 2012) investigated whether annotators followed guidelines, revealing potential cases of partiality. Additionally, (Mahamed Badel, 2020) demonstrated the use of a thesaurus for query expansion, where synonymous words are used to expand query terms. This approach is similar to a dictionary but lacks examples. Multilingual query translation, as experimented (Karakos et al., 2020; Boschee et al., 2019; Ogundepo et al., 2022) for information retrieval in low-resource languages like Somali, offers a promising corridor. Furthermore, (Vijayalakshmi and Dixit, 2019) explained how ontology can enhance query expansion, better than keyword-based systems. The sequential modules in an ontology-based retrieval system build on one another, as each module's output informs the next.

A technique proposed by (Yan and Gao, 2018) also entails diversifying the PRF retrieval process, in order to get more relevant documents and reduce overlooking user attention. However, PRF can be sensitive and ignores user attention, possibly causing some issues (Alhamed, 2019) introduced query modification using PRF, combining term expansion techniques to select important words for user queries. This study stated that the number of query terms doesn't matter, but sometimes can divert user expectations. In summary, however, there are various studies on both corpus construction and IR techniques. This study marks the first of its type for the Somali language IR evaluation.

Table 1: Somali IR evaluation corpus statistics.

| | |
|---|---|
| Number of sentences: | 16450 |
| Number of words: | 578216 |
| Number of documents: | 2335 |
| Average sentences per document: | 7.0449 |
| Average words per document: | 247.6269 |
| Average sentence length: | 35.1494 |
| Number of queries: | 16 |
| Average words per query: | 5.56 |

## 3 CORPUS

In this section of the paper, we briefly defined the construction process of the corpus and how we have annotated it. We also discussed the sources of the data collected. This section also introduces the query generation process and the technique applied to test the quality of the corpus.

### 3.1 Somali Language

The Somali language is a part of the Cushitic languages that form a branch of the Afro-Asiatic family [2]. The language is spoken in Africa, especially the Horn of Africa (Mohamed, 2013). The Somali language, as a member of the Cushitic family, is the second most spoken language in the Cushitic languages. Its writing system consists of 22 consonants, including a glottal stop. The Somali alphabet uses all alphabets in English except three (Z, V, and P). Symbols in Somali are (b, t, j, x, kh, d, r, s, sh, dh, c, g, f, q, k, l, m, n, w, h, y). In addition to this (Mohamed, 2013), the Somali language is an agglutinative language in terms of morphological formations.

### 3.2 Query Generation

During the corpus preparation, the first task we conducted was generating the queries. Like the task in (Esmaili et al., 2014), we have followed the standard test collection guidelines of TREC. Therefore, we here focus on three parts: documents, a set of queries, and a relevance judgment (Right answers).

Thus, to generate the queries or topics we directly interviewed a group of native speakers to forward their information needs based on subjects.

We have also asked them to provide a description of what the topics they are searching are about. Based on a set of guidelines that we gave them

---

[2]https://nalrc.indiana.edu/doc/brochures/somali.pdf

Table 2: Sample queries.

| Query | English Meaning |
|---|---|
| fatahaada wabiga shabeelle | Shabelle River floods |
| suuqa jigjiga ee gubtay | The burned Jigjiga mall |
| xaalada deegaanka tigreega | situation in tigray region |

to follow. Then, NLP researchers and linguists formulated the queries based on the information needed and the description provided by the information seekers. The guidelines set the length of the queries to be more than one word. In addition, the queries were also allocated to be from different categories of sources so that we could have a rich corpus. Moreover, the relevance of the queries was verified based on the documents in the corpus as discussed in section 3.4. After the information need of users is found and experts formulated the queries, we have shown sample relevant queries in Table 2.

### 3.3 Collecting the Documents

We have collected the corpus by following the approach used by most test collection studies such as (Haddow and Kirefu, 2020). Based on the queries generated in section 3.2, we manually collected news articles from several online Somali language posting sites, such as hiiraan online and dhacdo net news articles. Despite the manual collection of documents from hiiraan online and dhacdo net, we have also manually collected text from cultural and history books that are publicly available. One of such books is available at [3]. Because there can be some rights, we have asked permission to access these sites, as shown appendix B and C. The reason we chose target sources to be these sources is that in these sites and books, standard and error-free Somali words are found. The corpus annotation and cleaning are done by four researchers and two linguistics. The document is normalized by removing HTML tags, mentions, and languages other than Somali. A total of 2335 documents remain after we cleaned the documents. Figure 1 shows the size of the collected data among the different sources.

Moreover, Table 1 shows the statistics of the corpus such as the number of sentences, number of words, etc.

**Documents collected:** The annotated documents, are collected from various categories and

consist of domains such as sport, politics, culture, education, etc.

### 3.4 Relevance judgment

The decision that a query is relevant to a document in this study is made by annotators, who are language experts and NLP researchers. In accordance with the TREC test collection guidelines, a query can be regarded as relevant to a document if the query is found in any portion of the document. Furthermore, a document is considered relevant if it either contains the query terms or owns partial relevance to the query. In line with the guidelines, a document may also be considered relevant even if it lacks the exact query words, as long as it includes synonyms of the query terms.

**Nature of the queries used:** We have used several sets of queries to evaluate the effectiveness of the corpus. Our queries range from keyword-based queries to longer or more descriptive ones. We have also given consideration to the complexity of the queries using both simple and more complex queries. The queries span various topics in the corpus scope consisting of different user needs.

### 3.5 Labeling the documents and queries

Throughout the corpus annotation process, we assigned labels to both queries and documents. In total, we selected 16 queries or topics to judge against this corpus. These queries are denoted as $Q$-1, $Q$-2, $Q$-3, and so on up to $Q$-16, with the $Q$ denoting query terms. We also gave labels to the document ranging from $Som$-0000 to $Som$-2335.

Furthermore, besides giving labels to queries and documents, we created a set of judgment documents, that correspond to each query. The exact number of documents, we evaluated for every query are as follows: Queries $Q$-1 through $Q$-10 were assigned with 10 documents each, while $Q$-11 has 7, $Q$-12 has 5, $Q$-13 has 8, $Q$-14 has 5, $Q$-15 has 8, and $Q$-16 was judged against 11 documents. This makes 144 documents for the total relevant documents to the total queries.

### 3.6 Evaluating the Corpus

To evaluate the quality of the corpus, we have applied the pseudo-relevance feedback technique.

In this approach, the top $k$ documents are retrieved first, and from the top $k$ documents (Docs), top $m$ terms (top 5 terms in our case) are selected as expansion terms. A new query is refined (reformulated), and the refined query is given back

---

[3]https://maktabadda.com/diiwaanka-gabayadii-sayid-maxamed-cabdulle-xasan/diiwaanka-gabayadii-sayid-maxamed-cabdulle-xasan/

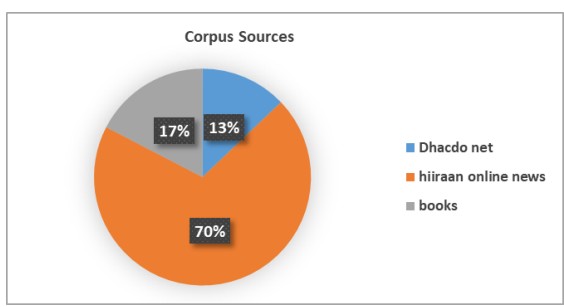

Figure 1: Sources of the corpus.

to the system to find more matching documents. The final documents are displayed to the user with the use of the refined query terms. PRF improves the retrieval performance by assuming that the $k$ highest-ranking documents in the first retrieval are relevant, and it selects query expansion terms from them and refines the query (Elvina and Mandala, 2020). The PRF technique does not need any interference from the user because the expansion terms are automatically added to the query. The queries, as well as the documents, are preprocessed initially. As we have discussed in section 3.7 of the paper Figure 2 shows how the Somali language information retrieval using PRF works. In collaboration with linguistics, we have identified a list of Somali language stop words. This stop-word identification was the first attempt, as far as we know. We then used this list of stop words and removed any word that matched with the stop words from the corpus during retrieval. These stop words are eliminated, from the text by reading a list of stop words in a text file and finding matches for them from the corpus.

After we have cleaned the document, tokenized it, and removed the stop words, we have created an inverted index. Lastly, documents are retrieved based on the new query, which is reformulated after the top 5 words are added to the initial query terms. In section 3.7, we implemented the following formulas to measure the performance.

$$P = \frac{\text{Retrieved Relevant}}{\text{total Retrieved}} \quad (1)$$

$$R = \frac{\text{retrieved Relevant}}{\text{total Relevant}} \quad (2)$$

$$F = \frac{2*PR}{P+R} \quad (3)$$

Where $P$ is the precision, $R$ is the recall, and $F$ is the f-measure. The *Retrieved Relevant* is the document retrieved and relevant to the queries.

The *total Retrieved* is the total number of the documents retrieved in response to the queries. The *total Relevant* is the number of documents that are relevant for a set of queries.

### 3.7 Experiment set up

In order to conduct the experiment, we have not used any framework, that implements the retrieval system. Rather we have applied a custom indexing system which is a Python-based tool. Our implemented program reads text documents from a specified directory, tokenizes the content into words, and then creates an inverted index for uni-grams from the content of those documents. The inverted index is essentially a data structure that associates each term with the documents in which it appears and the frequency of its occurrence within those documents. The indexing process in this study consists of creating an inverted index, calculating term frequency (TF) and document frequency (DF), sorting and writing token information, etc.

In Figure 2 we clearly show how we conducted this PRF query expansion retrieval process. Moreover, the development environment that we have used is the Pycharm editor.

## 4 RESULTS

For ranking the retrieved documents based on their relevance to the query, we used the best match 25 (BM25) ranking model (Zhao et al., 2018). The BM25 matching and ranking model sets values for parameters such as term frequencies, document frequency, and field length, etc.

One reason to use the BM25 model is that it is good in accuracy and fairness compared to other ranking models. Thus, utilizing this model, we have summarised the results in Table 3. The results are investigated with P@10, R@10, F@10, and P@20, R@20, F@20. Which literally means the performance of the system when the top 10 most relevant documents to the query are retrieved and when the top 20 most relevant documents to the query are retrieved. The proposed PRF approach is compared with TF-IDF to see the effect of the PRF on document retrieval over TF-IDF. As a result, the PRF technique outperformed the TF-IDF technique. By achieving an f-measure of 72% for F@10 and 57% for F@20 while, the TF-IDF scored an f-measure of 54% for F@10 and 44% for F@20. Finally, it shows that, in all cases, the PRF technique produced a good performance. Figure

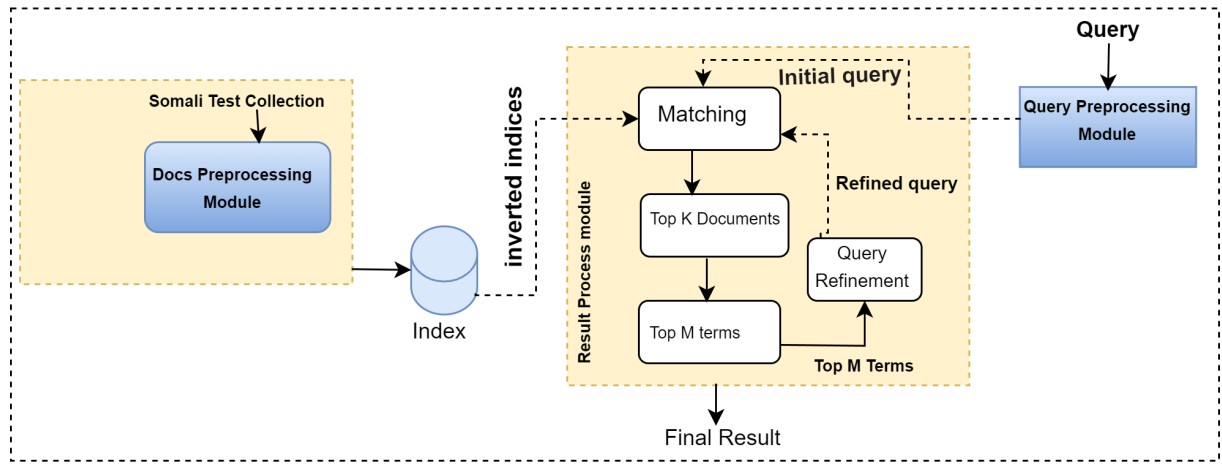

Figure 2: Retrieval process.

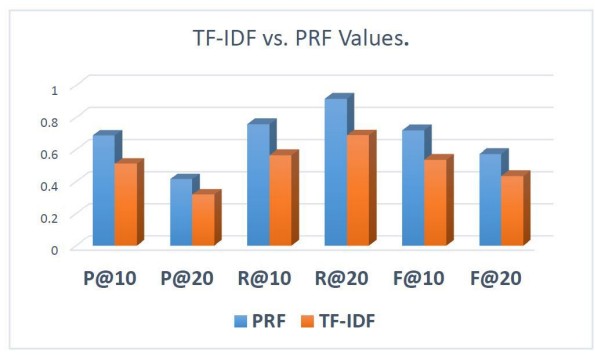

Figure 3: Comparison between PRF and TF-IDF.

Table 3: Results.

| Model | P@10 | P@20 | R@10 | R@20 | F@10 | F@20 |
|-------|------|------|------|------|------|------|
| TF-IDF: | 0.512 | 0.319 | 0.565 | 0.689 | 0.537 | 0.436 |
| **PRF:** | **0.687** | **0.415** | **0.758** | **0.917** | **0.721** | **0.572** |

3 shows the comparison between the PRF and the TF-IDF ranking models.

## 5 CONCLUSION AND FUTURE WORK

In this paper, we have introduced the creation of the first publicly available IR evaluation corpus for the Somali language. The corpus contains 2335 documents collected from various domains such as education, culture, etc. We have also provided a new list of Somali stop words and removed any stop word from the corpus using the created list. We then evaluated the corpus using the PRF query expansion technique, using well-established metrics like $P$, $R$, and $F$. The PRF technique exhibited its efficiency on the created corpus, achieving a recall of 92% for R@20. In comparison, the traditional TF-IDF ranking model scored lower compared to

PRF and scored a recall of 69% for R@20. This shows that the PRF technique outperforms TF-IDF in this corpus.

**Future work** In the future, we recommend the development of a fully functional stemmer for the Somali language, recognizing its significant usage in various NLP applications, including IR and machine learning. We would also like to extend this corpus to create a standard IR evaluation corpus for the Somali language.

## Limitations

We encountered a few limitations including a word in the Somali language may be written in different forms, for example, the Somali word $ra'iisulwasaare$ (Prime minister in English), may also be written as $raysulwasaare$ (Prime minister in English). Another limitation is that there is no fully functional stemmer for the Somali language, and this became a limitation of our study.

## Ethics Statement

This paper is an original work of the authors, it is not published in another place. It is also not currently being considered for publication in another place. The results reported are true and not falsified. All sources are properly cited and all researchers and co-authors actively contributed to this research.

## Acknowledgements

This work was supported in part by the National Natural Science Foundation of China under Grant 62072077 and Grant 62176043; in part by the Natural Science Foundation of Sichuan Province, China, under Grant 2022NSFSC0505; and in part by the

Kashgar Science and Technology Bureau under Grant KS2023025. *Hiiraan online*, *dhacdo net*, and the Somali *education bureau* are the main source providers. We extend our heartfelt appreciation to our esteemed language expert colleagues and fellow NLP researchers for their valuable contributions to this work. Additionally, we thank the three anonymous reviewers for their constructive comments.

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

## A  Sample Somali Stop words

Table 4:  Stop words.

| | | | | | |
|---|---|---|---|---|---|
| een | ooy | kaasi | ayuu | uuna | sii |
| ah | loo | sidii | iska | oo | kii |
| hore | isagoo | ay | kiina | sidee | sababtoo |
| hadii | weeye | yay | uusan | siisaa | bal |
| ayuu | kuwee | cid | inaga | uuna | kali |
| ayna | waxaan | kee | aanay | ah | kan |
| sidii | iska | ahayn | kii | ay | kiina |
| suu | naga | laakiin | iyo | kuwaa | wax |
| aynu | kuu | aan | kale | waxa | ayay |
| anaga | miyuu | ahaa | kaa | waa | ee |

## B  Dhacdo net text permission

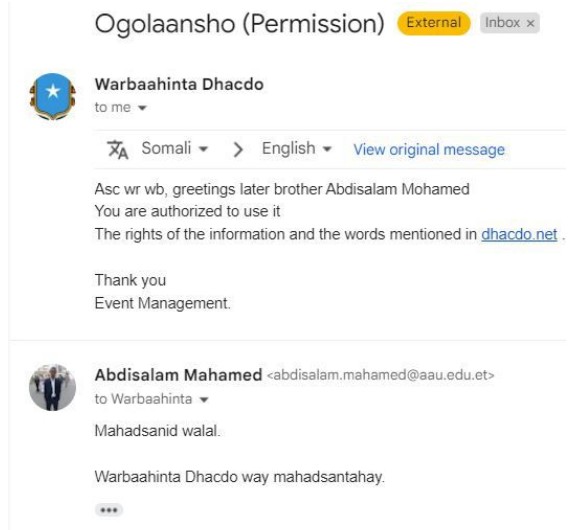

Figure 4:  Translated email of Dhacdo net.

## C  Hiiraan online text permission

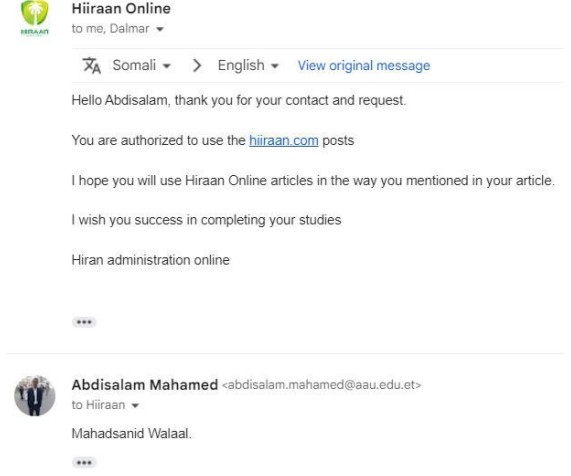

Figure 5:  Translated email of Hiiraan online.