# OpenReview forum: "Somali Information Retrieval Corpus: Bridging the Gap between Query Translation and Dedicated Language Resources"
_EMNLP/2023/Conference — EMNLP 2023 Main_

### Official Review · Reviewer_ReAG · 2023-08-03

**Soundness:** 3

**Excitement:**

4: Strong: This paper deepens the understanding of some phenomenon or lowers the barriers to an existing research direction.

**Missing References:**

The authors repeatedly refer to the existence of corpora for the
Somali language that are based on query translation, but no references
are provided.

**Paper Topic And Main Contributions:**

The authors present an (manually) annotated corpus for Somali language. The
contributions are in the field of the new data resources for
low-resource languages (Somali). The development of an information
retrieval system on this corpus has only a referential value in
relation to its quality.

**Questions For The Authors:**

A. Please, justify the authors' position in relation to the reviewer's
item B in the previous section "Reasons to reject".

The answer is adequate, but it would be necessary to review the new
version to issue a definitive opinion.

B. Please, justify the authors' position in relation to the reviewer's
item C in the previous section "Reasons to reject".

The answer is adequate, but the authors should include that
information in the revised version, and provide all the details
necessary to reproduce the tagging process.

C. Please, justify the authors' position in relation to the reviewer's
item D in the previous section "Reasons to reject".

The answer is adequate, but the authors should include that
information in the revised version, and also clarify how the concept
of "relevance" has been formalized.

D. Please, justify the authors' position in relation to the reviewer's
item E in the previous section "Reasons to reject".

The answer is confusing. Does not provide relevant information.

A. Please, justify the authors' position in relation to the reviewer's
item B in the previous section "Reasons to reject".

The answer is adequate, but it would be necessary to review the new
version to issue a definitive opinion.

B. Please, justify the authors' position in relation to the reviewer's
item C in the previous section "Reasons to reject".

The answer is adequate, but the authors should include that
information in the revised version, and provide all the details
necessary to reproduce the tagging process.

C. Please, justify the authors' position in relation to the reviewer's
item D in the previous section "Reasons to reject".

The answer is adequate, but the authors should include that
information in the revised version, and also clarify how the concept
of "relevance" has been formalized.

D. Please, justify the authors' position in relation to the reviewer's
item E in the previous section "Reasons to reject".

The answer is confusing. Does not provide relevant information.

A. Please, justify the authors' position in relation to the reviewer's
item B in the previous section "Reasons to reject".

The answer is adequate, but it would be necessary to review the new
version to issue a definitive opinion.

B. Please, justify the authors' position in relation to the reviewer's
item C in the previous section "Reasons to reject".

The answer is adequate, but the authors should include that
information in the revised version, and provide all the details
necessary to reproduce the tagging process.

C. Please, justify the authors' position in relation to the reviewer's
item D in the previous section "Reasons to reject".

The answer is adequate, but the authors should include that
information in the revised version, and also clarify how the concept
of "relevance" has been formalized.

D. Please, justify the authors' position in relation to the reviewer's
item E in the previous section "Reasons to reject".

The answer is confusing. Does not provide relevant information.

E. Section 5 cannot really be considered as a set of conclusions on the
quality of the elaborated corpus, but rather a simple summary of the
proposal. In this same line, the authors also fail to include a real
discussion of the practical results.

The answer is adequate.

**Reasons To Accept:**

Attempts to retrieve information in the Somali language are probably scarce, so
this work should contribute to alleviate this situation.

**Reasons To Reject:**

A. The writing style is very sloppy and includes, in addition to typos,
not a few grammatical constructions that seem out of context. The
perceived sensation is that sentences have been accumulated, without
paying too much attention to how they are linked together to form a
comprehensible text.

B. The only information on the size of the corpus is the number of
documents (2300), but we know nothing about the number of sentences,
the number of words or the average size of the sentences. The
information about the subject matter is also very scarce. Also
we don't know anything about the set of labels used.

C. The measures applied on the IR system used to evaluate the quality of
the corpus have some shortcomings. So, why the authors do not use
any of the standard measures, such as precision, recall, F-measure, ... ?

D. Also, in relation with this evaluation process, no data is provided
regarding the nature of the queries used.

E. Section 5 cannot really be considered as a set of conclusions on the
quality of the elaborated corpus, but rather a simple summary of the
proposal. In this same line, the authors also fail to include a real
discussion of the practical results.

**Reproducibility:**

4: Could mostly reproduce the results, but there may be some variation because of sample variance or minor variations in their interpretation of the protocol or method.

**Reviewer Confidence:**

4: Quite sure. I tried to check the important points carefully. It's unlikely, though conceivable, that I missed something that should affect my ratings.

**Typos Grammar Style And Presentation Improvements:**

As already mentioned, the wording is very sloppy and requires a
thorough revision. For example, Section 2 of the paper includes:

Line 095: "By conducting deep analysis authors" --> "By conducting deep analysis, authors"

Line 097: "What techniques have been used, what topics were hard are some of the issues studied in this paper." -->
     	  "What techniques have been used and what topics were hard are some of the issues studied in this paper."

Line 099: This sentence seems incomplete:

"To construct a corpus for information retrieval system that reflects real users (Kazai et al., 2012)."

Line 108: "Thesaurus lists words which are synonyms, it is similar to dictionary but lacks examples and definitions." -->

     	  "Thesaurus lists words which are synonyms, they are similar to dictionary but lacks examples and definitions."

Line 103: AS --> As

Line 117: "Ontology can also improve query expansion as shown by (Vijayalakshmi and Dixit, 2019)." -->
     	  "Ontologies can also improve query expansion as shown by (Vijayalakshmi and Dixit, 2019)."

Line 121: ontology based --> ontology-based

Line 124: This sentence seems incomplete:

"Novel approach that diversifies the pseudo relevance feed-back documents (Yan and Gao, 2018)."

All of these defects extend (to a lesser extent) to the rest of the
Sections. So, for example:

Line 057: "Annotated corpus is powerful tool for evaluation"  --> "Annotated corpus is a powerful tool for evaluation"

Line 058: information retrieval --> IR

Line 060: information retrieval --> IR

Line 064: "Text Retrieval Conference(TREC)" --> "Text Retrieval Conference (TREC)"

Line 174: "In the guideline length" --> "In the guideline, length"

Line 194: Judges --> judges

Line 216: "we have Applied" --> "we have applied"

Line 219: documents(Docs) --> documents (Docs)

Line 230: "The Queries as well as the documents are Preprocessed initially" --> "The queries as well as the documents are preprocessed initially"

Line 238: Missing punctuation marks in this sentence ?

"Stop words are eliminated from the text by reading a list of stop-
words in a text file any word which matches with the stop-words listed
in the file is then removed."

Line 242: "After the document is cleaned it is" --> "After the document is cleaned, it is"

Line 244: "reformulated(refined)" --> "reformulated (refined)"

Line 248: "documents(docs)" --> "documents (docs)"

Line 250: MB25 --> BM25

Line 265: "In addition to this Figure 3" --> "In addition to this, Figure 3"

Line 275: "poems and Proverbs" --> "poems and proverbs"

---

> ### Author Rebuttal · Authors · 2023-08-27
>
> We are very happy with your constructive comments those we think may help us enhance the writing style of our paper, We would like to give answers to some of the questions from the review the questions are numbered as A, B, C, D, E, Missing References, Typos Grammar Style And Presentation Improvements.
> ***
> > A. The writing style is very sloppy and includes, in addition to typos, not a few grammatical constructions that seem out of context. The perceived sensation is that sentences have been accumulated, without paying too much attention to how they are linked together to form a comprehensible text.
>
> Answer: We completely agree with you, that there are many styling problems in the paper. We will rewrite the whole text and submit a new version, which makes sense in the final, If our paper is accepted, the final version of the paper will be different than the current one, following the comments from the review.
> ***
>
> > B.  The only information on the size of the corpus is the number of documents (2300), but we know nothing about the number of sentences, the number of words or the average size of the sentences. The information about the subject matter is also very scarce. Also we don't know anything about the set of labels used.
>
> Answer: The corpus statistics, such as the number of sentences, the number of words, or the average size of the sentences is given as:
> * Number of sentences: **19320**
> * Number of words: **578170**
> * Number of documents: **2312**
> * Average sentences per document: **8.356401384083044**
> * Average sentence length: **29.927388469102578**
> * Average words per document: **250.0735294117647**
> * Number of queries: **16**
> * Average words per query: **4.9375**
>
>   **The information about the subject matter**: If your question is about the domain of the documents collected, it consists of domains such as sport, politics, culture, and education.
>
>     **Labels used**: The queries in this corpus are labeled as **Q-1**........**Q-16** and the documents are labeled as **Som-0000**.........**Som-2312**, we  hope that we have understood your concern.
> ***
>
> >  C. The measures applied on the IR system used to evaluate the quality of the corpus have some shortcomings. So, why the authors do not use any of the standard measures, such as precision, recall, F-measure, ... ?
>
> Answer: You are right, during the evaluation process of an IR system, the mentioned metrics make great sense. We would like to provide the  precision(P), recall(R) and F-measure(F) of the corpus by retrieving the top 10 and the top 20 most relevant documents to each query, the results are shown as:
>
> | metric    | **TF-IDF**|   &emsp;  &emsp;                                   **Pseudo relevance feedback (PRF) approach** |
> | -------- | ----------|-----------------------------------------------------------------------------------------------------|
> | P@10 | 0.506      |    &emsp;  &emsp;  &emsp;                                                              0.69                   |
> | R@10 | 0.562     |      &emsp;  &emsp;  &emsp;                                                        0.77                 |
> | F@10  | 0.532 |       &emsp;  &emsp;   &emsp;                                                           0.73                 |
> | P@20  | 0.31  |        &emsp;  &emsp;  &emsp;                                                      0.41                |
> | R@20  |  0.69 |       &emsp;  &emsp;   &emsp;                                                       0.92             |
> | F@20   | 0.43 |       &emsp;  &emsp;  &emsp;                                                           0.57               |
> ---------------------------------------------------------------------------------------------------------------------------------
>
> To calculate the above metrics we have used the following formulas for precision, recall, and f-measure:
>
> **P** = Retrieved relevant/total Retrieved
>
> **R** = retrieved relevant/total relevant
>
> **F**= (2*PR)/P+R
>
> Where R is the recall, P is the precision and F is the F-measure.
> ***
> > D. Also, in relation to this evaluation process, no data is provided regarding the nature of the queries used.
>
> Answer: The nature of the queries that we have used in the evaluation of this corpus, span different categories such as retrieving information about prominent players, culture, historical sites, songs, and educational information, for example, Q-6: wabiga shabeelle (Shabelle river).
> ***
> > E. Section 5 cannot really be considered as a set of conclusions on the quality of the elaborated corpus, but rather a simple summary of the proposal. In this same line, the authors also fail to include a real discussion of the practical results.
>
> Answer: We would like to express our keenness to re-arrange the conclusion part of the paper improve it, and add some practical aspects of the corpus.
> ***
> > Missing References: The authors repeatedly refer to the existence of corpora for the Somali language that is based on query translation, but no references are provided.
>
> Answer: We are too concerned about not citing referenced authors in the places you mentioned, but we were referencing (Karakos2020, Boschee2019) in the related work of the paper.
> ***
> > Typos Grammar Style And Presentation Improvements: As already mentioned, the wording is very sloppy and requires a thorough revision. For example, Section 2 of the paper includes:
>
> Answer: Indeed, we see the errors you figured out in the paper, section 2, and also other parts of the paper. The final version of the paper will not look like this if it is accepted at the conference. We will do a profound proofreading and improve any error you figured out in the paper.
> ***
>  We appreciate your questions and hope these responses address your concerns.
> ****

---

### Official Review · Reviewer_PkYX · 2023-08-04

**Soundness:** 3

**Excitement:**

4: Strong: This paper deepens the understanding of some phenomenon or lowers the barriers to an existing research direction.

**Paper Topic And Main Contributions:**

The paper presents a new dataset for Information Retrieval in Somali. The authors created a corpus of 2K documents and a set of queries. Then documents were labeled as relevant or not by annotators.

The annotation and data collection was done by native speakers and linguists, and is relevant due to the lack of NLP resources for Somali. The authors also point out the lack of proper stemmer for Somali and created a list of stop words.

**Questions For The Authors:**

Can you please provide the data mentioned in the previous section? If authors are able to provide them it would improve the overall Soundness of the paper, and I would reconsider its Soundness score.

**Reasons To Accept:**

- Somali is spoken by more than 20 million people, however resources are scarce. A new IR dataset is a small but important contribution to closing the gap with other languages. As comparison, languages such as Swedish, Finish or Bulgarian are spoken by far less speakers but have many more systems and resources available and have stemmers.

- The dataset was created by native speakers and linguists with knowledge of the language.

- They provide a new list of stop words for Somali.

**Reasons To Reject:**

- The paper still needs work in some areas. It lacks certain details that are important when presenting a new resource. I would suggest authors provide with complete numbers, as the total number of queries in the dataset, and perhaps some statistics, such as the average length of documents and queries. The Figures with results are too fine-grained. While they show how the proposed baseline has some improvement over tf-idf, it is done at the sample level for just 10 queries. I think there's a need for a table with results over the total dataset, by providing some relevant metric such as Recall@10 or similar.

**Reproducibility:**

4: Could mostly reproduce the results, but there may be some variation because of sample variance or minor variations in their interpretation of the protocol or method.

**Reviewer Confidence:**

4: Quite sure. I tried to check the important points carefully. It's unlikely, though conceivable, that I missed something that should affect my ratings.

**Typos Grammar Style And Presentation Improvements:**

The current version of the paper needs improvement on the use of English. I suggest to share the paper with an English native speaker to check it, and if that is not possible, consider using a tool like Grammarly to correct it.

---

> ### Author Rebuttal · Authors · 2023-08-27
>
> We appreciate your helpful feedback on the review of our paper. We would like to provide answers to specific questions below.
> ***
>
> > Q1: The paper still needs work in some areas. It lacks certain details that are important when presenting a new resource. I would suggest authors provide with complete numbers, as the total number of queries in the dataset, and perhaps some statistics, such as the average length of documents and queries.
>
> Answer: Right, the statistics are important information about a paper presenting a new resource. Therefore, below we have discussed the above question and listed the statistics of the corpus.
>
>  * The number of queries in the data set is a total of **16** queries.
> *  Number of sentences: **19320**
> * Number of words: **578170**
> *  Number of documents: **2312**
> *  Average sentences per document: **8.356401384083044**
> *  Average sentence length: **29.927388469102578**
> *  Average words per document: **250.0735294117647**
> * Number of queries: **16**
> *  Average words per query: **4.9375**
>
> ****
> > Q2: The Figures with results are too fine-grained. While they show how the proposed baseline has some improvement over tf-idf, it is done at the sample level for just 10 queries. I think there's a need for a table with results over the total dataset, by providing some relevant metric such as Recall@10 or similar.
>
> Answer:  To answer the above question we have computed precision(P), recall(R) and f-measure(F) for the queries across the entire dataset, The performance is tested by retrieving the top 10 and the top 20 most relevant documents(k) to the queries, we have discussed precision@10, recall@10, precision@20, recall@20, f-measure@10 and f-measure@20 as:
>
>
> | metric    | **TF-IDF**|             &emsp;   &emsp;                        **Pseudo relevance feedback (PRF) approach** |
> | --------  |  ---------- | -----------------------------------------------------------------------------------------------------|
> | P@10  | 0.506       |   &emsp;      &emsp;   &emsp;                                                  0.69                                    |
> | R@10  | 0.562        |    &emsp;        &emsp;  &emsp;                                                   0.77                            |
> | F@10   | 0.532 |      &emsp;      &emsp;    &emsp;                                                       0.73                     |
> | P@20   | 0.31  |       &emsp;        &emsp;   &emsp;                                                0.41                |
> | R@20   |  0.69 |        &emsp;         &emsp;   &emsp;                                              0.92             |
> | F@20    | 0.43 |         &emsp;        &emsp;    &emsp;                                                 0.57               |
> ---------------------------------------------------------------------------------------------------------------------------------
>
> To calculate the above metrics we have used the following formulas for precision, recall, and f-measure:
>
> **P** = Retrieved relevant/total Retrieved
>
> **R** = Retrieved Relevant/total Relevant
>
> **F**= (2*PR)/P+R
>
> Where R is the recall, P is the precision and F is the F-measure.
>
> ***
> > Q3: The current version of the paper needs improvement on the use of English. I suggest to share the paper with an English native speaker to check it, and if that is not possible, consider using a tool like Grammarly to correct it.
>
> Answer: We understand that the current version of the paper has various grammatical errors, so we take this comment and improve the writing style of the final version of the paper.
> ****
> We believe that we have tried to address and give answers to your comments.
> ****

---

### Official Review · Reviewer_gWGd · 2023-08-05

**Typos Grammar Style And Presentation Improvements:** 099-101
**Soundness:** 4

**Excitement:**

4: Strong: This paper deepens the understanding of some phenomenon or lowers the barriers to an existing research direction.

**Paper Topic And Main Contributions:**

This short paper presents the creation of a TREC-like Information Retrieval corpus for the Somali language. It presents the collection of queries from users, the crawling of a corpus and the generation of reference document lists from the corpus for each query. A secondary result is the first list of stop words for Somali.

**Reasons To Accept:**

Creating linguistic resources for low resource languages is a very important topic, particularly for languages from families with few high resources languages.
The work presented seems to be correctly made using classical methods, reproducing techniques initially created for English. This will allow to compare future results with state of the art.

**Reasons To Reject:**

The corpus is crawled from online resources, and it will be made available. But the question of text copyrights is not stated at all. If the authors have not obtained the necessary authorizations, the main result will be illegal. This must absolutely be cleared up before publication. From authors answer, this will be done for final version.

The number of documents manually judged to define the sets of valid documents for each query is not clear. Similarly, the indexing and matching tools are not explicit. Authors answer clear up these questions. I Suggest for future works to use an indexing engine like Elastic Search or another to avoid having to reproduce everything.

**Reproducibility:**

4: Could mostly reproduce the results, but there may be some variation because of sample variance or minor variations in their interpretation of the protocol or method.

**Reviewer Confidence:**

3: Pretty sure, but there's a chance I missed something. Although I have a good feel for this area in general, I did not carefully check the paper's details, e.g., the math, experimental design, or novelty.

---

> ### Author Rebuttal · Authors · 2023-08-27
>
> Thank you for the detailed review and thoughtful feedback. In the next lines of text, we focus on answering some questions.
> ***
>
> > Q1: The corpus is crawled from online resources, and it will be made available. But the question of text copyrights is not stated at all. If the authors have not obtained the necessary authorizations, the main result will be illegal. This must absolutely be cleared up before publication.
>
> Answer: It's crucial to highlight that our process of compiling the corpus was extremely thorough, taking into account the usage terms of the websites we accessed.  We collected the text from various websites and annotated only those we had permission to use. Therefore we would like to add the legal authorization to the final paper appendix.
> ***
> >Q2: The number of documents manually judged to define the sets of valid documents for each query is not clear. Similarly, the indexing and matching tools are not explicit.
>
> Answer: Yes, in the submitted version of the paper we have not added such information, we regret not adding such a piece of important information, here is the answer to your question:
>
>  1). The number of documents manually judged to define the sets of valid documents for each query(Q) are discussed as:
>      Q-1.....Q-10 (Q-1 up to Q-10) has 10 valid documents each, Q-11 has 7 valid documents, Q-12 has 5 valid documents, Q-13 has 8 valid documents, Q-14 has 5 valid documents, Q-15 has 8 valid documents and Q-16 has 11 valid documents.
>
> 2). The indexing Process we have applied:
>
> To create the indexing we have utilized a custom indexing system using a Python-based tool, our program reads text documents from a specified directory, tokenizes the content into words, and then creates an inverted index for unigrams (single words) from the content of those documents.
>
> Our main indexing process involves:
>
> * An inverted Index is created
> *  Term Frequency (TF) and Document Frequency (DF) are calculated.
> *  We sort and write to files.
> *  Tokens information is then written.
>
> 3). The matching and scoring tool adopted in this paper is the BM25 (best match 25), which is a probabilistic algorithm.
> ***
> >Q3: 099-101: sentence meaning is hard to get More generally, Section 2 is hard to follow as there is a lack of connections between its sentences. I am myself, not a native English speaker, but I have the feeling that this paper could be largely improved using a proofreading by a native.
>
> Answer: We agree with you that our paper contains many grammatical errors. We would like to revise Section 2 and produce a well-organized section in the final version.
>
> We hope, that we have at least attempted to give answers to your questions.
> ***

---

### Meta-Review · Area_Chair_XN9c · 2023-09-07

**Recommendation:** 4

**Metareview:**

This paper introduces a novel information retrieval corpus as well as a list of stop words for Somali, thus addressing a gap in NLP research for a low-resource language. While the excitement and potential impact of this work deserves praise, some details are vague or missing, e.g. the total number of queries in the dataset, statistics regarding the length of the documents and queries, etc. These concerns were nevertheless addressed by the authors in their rebuttals, and reviewers acknowledged these responses as satisfactory. There is, however, substantiated concern over style and presentation, suggesting the paper would benefit from the input of a native speaker or at minimum, an automatic grammar checker like Grammarly.

In summary, the revisions required to address paper content concerns can be accomplished quite readily, as indicated by fruitful discussions during the rebuttal period. The bulk of revisions would likely be dedicated to ensuring the writing is clear and up to standard.

---

### Decision · Program_Chairs · 2023-10-07

**Decision:**

Accept-Main

**Comment:**

This paper introduces a novel information retrieval corpus as well as a list of stop words for Somali, thus addressing a gap in NLP research for a low-resource language. While the excitement and potential impact of this work deserves praise, some details are vague or missing, e.g. the total number of queries in the dataset, statistics regarding the length of the documents and queries, etc. These concerns were nevertheless addressed by the authors in their rebuttals, and reviewers acknowledged these responses as satisfactory. There is, however, substantiated concern over style and presentation, suggesting the paper would benefit from the input of a native speaker or at minimum, an automatic grammar checker like Grammarly.

In summary, the revisions required to address paper content concerns can be accomplished quite readily, as indicated by fruitful discussions during the rebuttal period. The bulk of revisions would likely be dedicated to ensuring the writing is clear and up to standard.